# Multivariate discovery and replication of five novel loci associated with Immunoglobulin G *N*-glycosylation

Xia Shen [1,2,3], Lucija Klarić [1,3,4], Sodbo Sharapov[5,6], Massimo Mangino [7,8], Zheng Ning[2], Di Wu[9], Irena Trbojević-Akmačić[4], Maja Pučić-Baković[4], Igor Rudan[1,3], Ozren Polašek[10], Caroline Hayward [3], Timothy D. Spector[7], James F. Wilson [1,3], Gordan Lauc[4,11] & Yurii S. Aulchenko [5,6,12]

Joint modeling of a number of phenotypes using multivariate methods has often been neglected in genome-wide association studies and if used, replication has not been sought. Modern omics technologies allow characterization of functional phenomena using a large number of related phenotype measures, which can benefit from such joint analysis. Here, we report a multivariate genome-wide association studies of 23 immunoglobulin G (IgG) *N*-glycosylation phenotypes. In the discovery cohort, our multi-phenotype method uncovers ten genome-wide significant loci, of which five are novel (*IGH, ELL2, HLA-B-C, AZI1, FUT6-FUT3*). We convincingly replicate all novel loci via multivariate tests. We show that IgG *N*-glycosylation loci are strongly enriched for genes expressed in the immune system, in particular antibody-producing cells and B lymphocytes. We empirically demonstrate the efficacy of multivariate methods to discover novel, reproducible pleiotropic effects.

[1] Centre for Global Health Research, Usher Institute of Population Health Sciences and Informatics, University of Edinburgh, Teviot Place, Edinburgh EH8 9AG, Scotland, UK. [2] Department of Medical Epidemiology and Biostatistics, Karolinska Institutet, Nobels väg 12 A, SE-17 177 Stockholm, Sweden. [3] MRC Human Genetics Unit, MRC Institute of Genetics and Molecular Medicine, University of Edinburgh, Western General Hospital, Crew Road, Edinburgh EH4 2XU, Scotland, UK. [4] Genos Glycoscience Research Laboratory, Hondlova 2/11, Zagreb 10000, Croatia. [5] Novosibirsk State University, Pirogova 2, Novosibirsk 630090, Russia. [6] Institute of Cytology and Genetics SB RAS, Lavrentyeva ave. 10, Novosibirsk 630090, Russia. [7] Department for Twin Research, King's College London, London WC2R 2LS, England, UK. [8] National Institute for Health Research (NIHR) Biomedical Research Centre at Guy's and St. Thomas' Foundation Trust, London SE1 9RT, England, UK. [9] Science for Life Laboratory, Department of Biochemistry and Biophysics, Stockholm University, Tomtebodavägen 23B, Stockholm SE-171 65, Sweden. [10] Faculty of Medicine, University of Split, Šoltanska ul. 2, Split 21000, Croatia. [11] Faculty of Pharmacy and Biochemistry, University of Zagreb, A. Kovacica 1, Zagreb 10000, Croatia. [12] PolyOmica, Het Vlaggeschip 61, 's-Hertogenbosch 5237PA, The Netherlands. Correspondence and requests for materials should be addressed to X.S. (email: xia.shen@ed.ac.uk) or to Y.S.A. (email: y.s.aulchenko@polyomica.com)

A large number of genetic loci have been mapped for complex traits of clinical relevance via genome-wide association studies (GWAS). However, our understanding of the biology behind the association is rather limited for most of these loci. In order to move further towards the goal of unraveling the biological mechanisms underpinning complex disease, large-scale omics data have been generated as intermediate phenotypes to help fill in the gap between the genome and organismal level traits. Nonetheless, in contrast to the complex traits that are commonly analyzed, the large number of omics variables often have a strong correlation structure that cannot be neglected[1]. Therefore, instead of conventional univariate GWAS, multivariate analysis approaches are needed in GWA analyses to incorporate the ubiquitous, partly genetically regulated, correlation among omics variables.

Many multivariate methods have been developed for genetic analyses[2, 3] (see Supplementary Note 1 for details, also review by ref. [4]). In GWAS context, multivariate omics analysis with explicit modelling of phenotypic covariance faces several challenges. In genetically structured samples, this structure needs to be taken into account. The multivariate methods allowing for covariates can handle population stratification by including principle components of the genomic kinship matrix in the model. However, the effects of small groups of close relatives—if present—are usually not reflected in the leading principal components because such kinship generates weaker LD than large-scale population structure does[5]. In such situations, the method of choice is mixed effect models; however, a fast mixed model analysis for multiple (more than tens of) phenotypes is still difficult. GEMMA[6] and Limix[7] were designed to overcome such difficulty, although the computational cost can be high for a large number of phenotypes such as that encountered in omics data. Another difficulty faced in multivariate analysis of omics phenotypes is finding a method for replication, which would allow for both tests of significance and consistency of the model being replicated.

Despite the number of methodological studies, few empirical multivariate GWAS have been published for humans. This may be largely attributed to the statistical complexity and computational difficulty of such analyses as well as outlined above issues addressing population stratification, replication, and interpretation. Among multivariate GWAS conducted to date, most are dealing with very complex phenotypes (e.g., refs [8, 9]). These studies usually propose a new method, and investigate real data to demonstrate that one may expect increased mapping power when using these. However, few new loci have been convincingly identified using such methods for complex phenotypes, and no replication was attempted. Therefore—at least for the moment—the practical gain from multivariate GWAS of complex traits seems to be rather limited. The situation appears to be somewhat different for traits more proximal to the genotype. Stephens[10] applied multivariate GWAS to summary level data from the Global Lipids Genetics Consortium (GLGC-2010[11], N up to 100,184) and identified 18 new loci on the top of 95 loci identified in single trait analyses. While no replication was provided in this work, importantly, we observe that 11 of the loci identified by multivariate analyses were later confirmed with genome-wide significance in the next, bigger, univariate meta-analysis by GLGC in 2013[12] (N up to 188,577). It appears the yield of new loci is even bigger for metabolomics traits. Inouye et al.[13] performed multivariate GWAS of 130 NMR metabolites (grouped in 11 sets) in 6600 individuals. The study demonstrated that multivariate analysis doubles the number of loci detected in this sample. Again, no replication was performed, however, among seven novel loci discovered by Inouye et al. via multivariate analysis, three demonstrate genome-wide significant ($P < 5 \times 10^{-11}$)

association with at least one NMR metabolite investigated recently in a sample of up to 24,925 individuals[14].

These observations strongly suggest that at least some of the novel loci discovered via multivariate GWAS are true positives. The fact that "univariate" replication of novel loci from the lipid and metabolomic studies is not perfect (around 50%), even after substantially increased sample size, suggests that either multivariate analyses are more prone to the appearance of statistical artifacts and/or are subject to different genome-wide significance thresholds, or that the underlying pleiotropic effects captured in multivariate models are hard to capture using standard univariate analyses (see ref. [10] for examples of such scenarios), and require different approach to replication.

Here, we explore the potential of multivariate methods for studying genetic regulation of glycan variation. Glycans are complex carbohydrates bound to the surface of proteins, whose structure and function they consequently substantially influence. Glycosylation is one of the most abundant post-translational protein modifications[15, 16], but knowledge about its biological function was long hindered by glycans' structural complexity. Unraveling the complex network of genes involved in protein glycosylation can provide not only a better understanding of this fundamental biological process, but might also provide insights into how these molecules could be involved in complex human diseases, and potentially used as biomarkers in prediction of disease susceptibility[17–20].

Previous GWAS have analyzed either the N-glycans released from all plasma proteins[21, 22] or focused on N-glycosylation of a single protein—Immunoglobulin G (IgG), which is first isolated from other plasma proteins, followed by quantification of enzymatically-released glycans[23]. These studies uncovered six loci for total plasma and nine loci for IgG glycosylation[21–23]; with only one locus (FUT8) overlapping between the two. The majority of plasma proteins are synthesized in the liver and pancreas[24] while immunoglobulins are synthesized specifically in cells of the immune system[25]. The lack of overlap between results of these genetic studies suggests different mechanisms of biological control of glycosylation in these two tissues.

Here, we apply multivariate methods to the IgG N-glycosylation traits and empirically test whether novel loci can be convincingly replicated. To utilize data from populations with high kinship, we formulate and implement a multivariate GWAS workflow based on combination of a linear-mixed-model-based phenotypic transformation, MANOVA, and multiple regression.

## Results

**Joint analysis of IgG glycans identifies five novel loci.** A single protein (IgG) was isolated from the plasma of 1960 individuals from the population of the Orkney Islands in Northern Scotland (the ORCADES cohort[26]). The N-linked glycans were assayed using ultra performance liquid chromatography (UPLC), resulting in 23 quantitative measurements.

The heart of our analysis procedure is, essentially, MANOVA statistics (see Methods for details). However, in order to avoid the effect of confounding by population genetic structure and kinship, prior to MANOVA analysis each phenotype undergoes linear-mixed-model-based GRAMMAR+ transformation we have previously developed[27] as an improvement of GRAMMAR procedure[28] for rapid association analysis in pedigrees and samples form genetically isolated populations.

We performed multivariate GWAS combining all the 23 IgG N-glycosylation phenotypes using MANOVA. The same procedure was also applied to eight subsets of the traits based on different chemical and structural properties of glycans, namely galactosylation, monogalactosylation, digalactosylation,

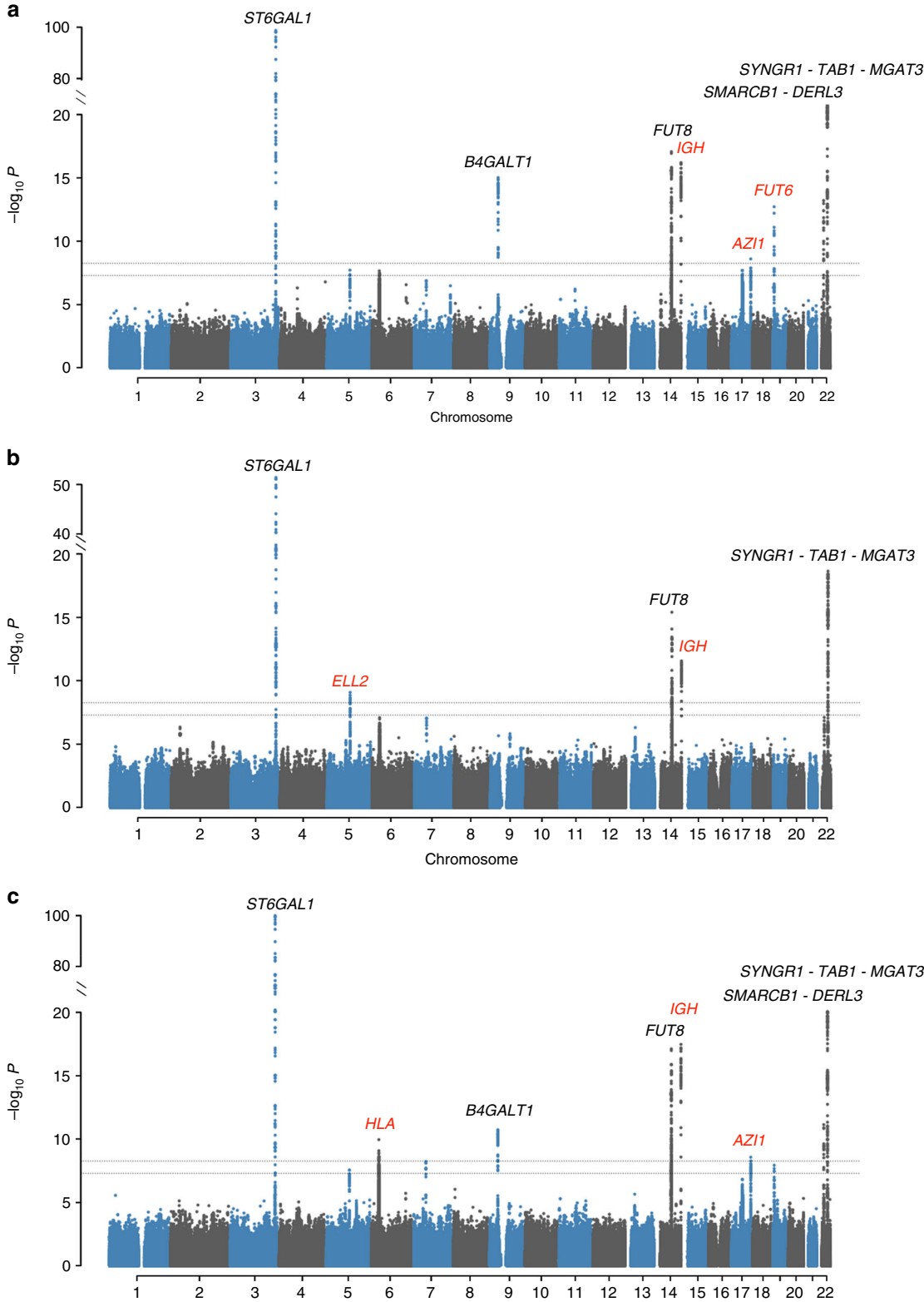

**Fig. 1** Manhattan plots of multivariate GWAS for IgG *N*-glycosylation phenotypes in the ORCADES discovery population. The known and novel loci are labeled in *black* and *red*, respectively. 23 IgG *N*-glycosylation phenotypes were analyzed together and also in eight different functional subgroups, including sialylation and galactosylation. **a** Analysis of 23 *N*-glycosylation traits; **b** Analysis of eight sialylation phenotypes; **c** Analysis of 17 galactosylation phenotypes. The *horizontal dashed lines* represent the genome-wide significant *P*-value threshold of $5\times10^{-8}/9 = 5.6\times10^{-9}$ and the genome-wide suggestive significant threshold of $5\times10^{-8}$

**Table 1 Novel loci detected via multivariate GWAS for IgG *N*-glycosylation phenotypes**

| Candidate genes | Phenotypes grouping | Top variant | EA | Discovery: ORCADES | | Replication I: KORCULA + VIS | | | | Replication II: TWINSUK | | | |
|---|---|---|---|---|---|---|---|---|---|---|---|---|---|
| | | | | $\beta_S$ (s.e.) | $P$ | $P_{MANOVA}$ | $\beta_S$ (s.e.) | $P_S$ | $r_\rho$ | $P_{MANOVA}$ | $\beta_S$ (s.e.) | $P_S$ | $r_\rho$ |
| ELL2 | digalactosylation | rs11135441 | T | 0.0336 (0.0056) | 1.6E−09 | 4.5E−04 | 0.0276 (0.0058) | 2.1E−06 | 0.60 | 6.9E−09 | 0.0183 (0.0027) | 5.5E−12 | 0.86 |
| | sialylation | rs11135441 | T | 0.0313 (0.0050) | 5.0E−10 | 1.1E−04 | 0.0234 (0.0056) | 2.7E−05 | 0.76 | 3.6E−09 | 0.0091 (0.0014) | 2.7E−10 | 0.96 |
| HLA-B-C | galactosylation | rs116108880 | G | 0.0408 (0.0066) | 8.3E−10 | 6.2E−01 | 0.0188 (0.0053) | 4.0E−04 | 0.39 | 3.8E−08 | 0.0160 (0.0031) | 2.2E−07 | 0.74 |
| IGH | N-glycosylation | rs35590487 | T | 0.0671 (0.0080) | 6.3E−17 | 2.3E−02 | 0.0218 (0.0074) | 3.1E−03 | 0.10 | 1.0E−31 | 0.0534 (0.0040) | 5.2E−41 | 0.76 |
| | monogalactosylation | rs35590487 | T | 0.0465 (0.0055) | 3.6E−17 | 7.1E−02 | 0.0178 (0.0040) | 9.5E−06 | 0.90 | 6.3E−25 | 0.0358 (0.0032) | 1.2E−28 | 0.86 |
| | galactosylation | rs35590487 | T | 0.0637 (0.0073) | 3.4E−18 | 8.8E−02 | 0.0212 (0.0059) | 3.3E−04 | 0.02 | 4.7E−30 | 0.0501 (0.0039) | 4.1E−38 | 0.85 |
| | monosialylation | rs35590487 | T | 0.0325 (0.0046) | 9.9E−13 | 3.5E−01 | 0.0114 (0.0044) | 9.9E−03 | 0.99 | 6.3E−24 | 0.0287 (0.0026) | 1.2E−27 | 0.97 |
| | sialylation | rs35590487 | T | 0.0379 (0.0054) | 1.5E−12 | 3.5E−01 | 0.0113 (0.0050) | 2.3E−02 | 0.36 | 5.9E−23 | 0.0287 (0.0026) | 1.2E−27 | 0.94 |
| | fucosylation | rs58087925 | T | 0.0625 (0.0070) | 3.8E−19 | 7.2E−02 | 0.0242 (0.0059) | 4.5E−05 | 0.18 | 3.7E−34 | 0.0523 (0.0038) | 8.3E−43 | 0.93 |
| | bisecting GlcNAc | rs8013055 | A | 0.0370 (0.0052) | 1.0E−12 | 3.0E−01 | 0.0077 (0.0058) | 1.8E−01 | 0.70 | 1.1E−15 | 0.0261 (0.0029) | 1.9E−19 | 0.96 |
| AZI1 | N-glycosylation | rs9319617 | C | 0.0453 (0.0076) | 2.5E−09 | 9.7E−03 | 0.0204 (0.0064) | 1.4E−03 | 0.69 | 7.6E−07 | 0.0199 (0.0033) | 2.2E−09 | 0.75 |
| | galactosylation | rs9319617 | C | 0.0422 (0.0073) | 2.7E−09 | 2.6E−03 | 0.0189 (0.0047) | 6.2E−−05 | 0.79 | 1.2E−06 | 0.0176 (0.0029) | 8.5E−10 | 0.78 |
| | fucosylation | rs2659009 | A | 0.0414 (0.0063) | 3.9E−11 | 5.1E−04 | 0.0168 (0.0049) | 5.8E−04 | 0.85 | 6.0E−05 | 0.0159 (0.0027) | 4.0E−09 | 0.89 |
| FUT6-3 | N-glycosylation | rs12019136 | A | 0.0574 (0.0078) | 1.9E−13 | 3.3E−01 | 0.0022 (0.0070) | 7.5E−01 | 0.01 | 3.3E−15 | 0.0251 (0.0031) | 6.3E−16 | 0.82 |

Nine multivariate GWA scans were performed, including one using all the 23 phenotypes, as well as eight different subgroupings according to type of glycosylation. Replication was performed by (i) MANOVA test in the replication cohorts ($P_{MANOVA}$); and (ii) testing the association between the phenotypic score (constructed based on the coefficients estimated in the discovery cohort) and the corresponding genotype dosages ($P_S$, reported only for replication cohorts). $\beta_S$ denotes the coefficient of regression of genotype dosage onto the phenotypic score. Consistency of effects was performed by testing the correlation of partial genotype-phenotype correlations ($r_\rho$) in the discovery and replication cohorts. EA, effect allele. Extended details are given in Supplementary Table 6.

sialylation, monosialylation, disialylation, fucosylation and bisecting GlcNAc (*N*-acetyl glucosamine). The definition of subgroups is given in Supplementary Table 2. The distribution of the observed test statistic was contrasted to that expected under the null hypothesis, similar to the genomic control[29] (Supplementary Fig. 1). Given that we have analyzed nine groups of traits, although they are strongly overlapping, we decided to take a conservative approach and have considered nominal MANOVA *P*-values < $(5 \times 10^{-8})/9 = 5.6 \times 10^{-9}$, as genome-wide significant.

In total, nine GWA scans were performed. Our multivariate analysis replicated five out of nine previously established[23] IgG *N*-glycosylation loci (*ST6GAL1, B4GALT1, FUT8, SMARCB1-DERL3* and *SYNGR1-TAB1-MGAT3*; Fig. 1, Supplementary Table 3) and detected five new loci (*IGH, ELL2, HLA-B-C, AZI1, FUT6-FUT3*, named by genes according to functional candidacy; Table 1, Supplementary Table 6, Supplementary Data 1). Among the five novel loci, four were not detectable using the conventional univariate analysis. The *IGH* locus could be identified using univariate GWAS on GP9, however, it was not mapped in the previous univariate GWAS with even larger discovery sample[23] (N = 2247).

**Genetic effects on phenotype scores are strongly replicated**. To replicate and interpret the genetic effect of each newly identified locus (Table 1, Supplementary Table 6), one could perform a multivariate test, resulting in a *P*-value, which, when significant, might be interpreted as replication. However, in regular single-trait GWAS, a stronger replication criterion is used. This criterion asks for both significance and consistency in the direction of effect, i.e., that the same allele should be associated with increased/decreased risk or trait value in both discovery and replication samples. To implement such a 'significant and consistent' replication criterion, we first suggest estimating a linear combination of the phenotypes, i.e., constructing a phenotypic score, S, that best fits the associated genotype in the discovery cohort, and then computing the score and testing its association with the genotype in a replication cohort. If the same allele is used as reference in both discovery and replication cohorts, this consistency criterion translates into finding a positive association between the phenotypic score and the genotype in the replication cohort. In Table 1, $\beta_S$ and $P_S$ represent the estimation and replication of the genotypic effects on such phenotype scores. The coefficients to construct the scores were estimated using a linear regression of the SNP dosage on the phenotypes (Supplementary Table 4). For each locus, the same score, with coefficients estimated only in the discovery cohort, was tested against the same variant in two independent Croatian cohorts, KORCULA (n = 850) and VIS (n = 840), and meta-analyzed for replication. Following the same protocol, we performed a second replication in a much larger cohort TWINSUK (n = 4479), which confirmed all novel loci at a high significance level (Table 1; regression *t*-test $P < 0.05/5/9 = 1.1 \times 10^{-3}$, are considered significant in replication). Such estimates also allow us to perform meta-analysis of estimates from individual cohorts, resulting in high significance for all the five newly discovered loci (Supplementary Table 6).

To be confident about our replication results and to verify the suggested replication procedure, we also considered "univariate"

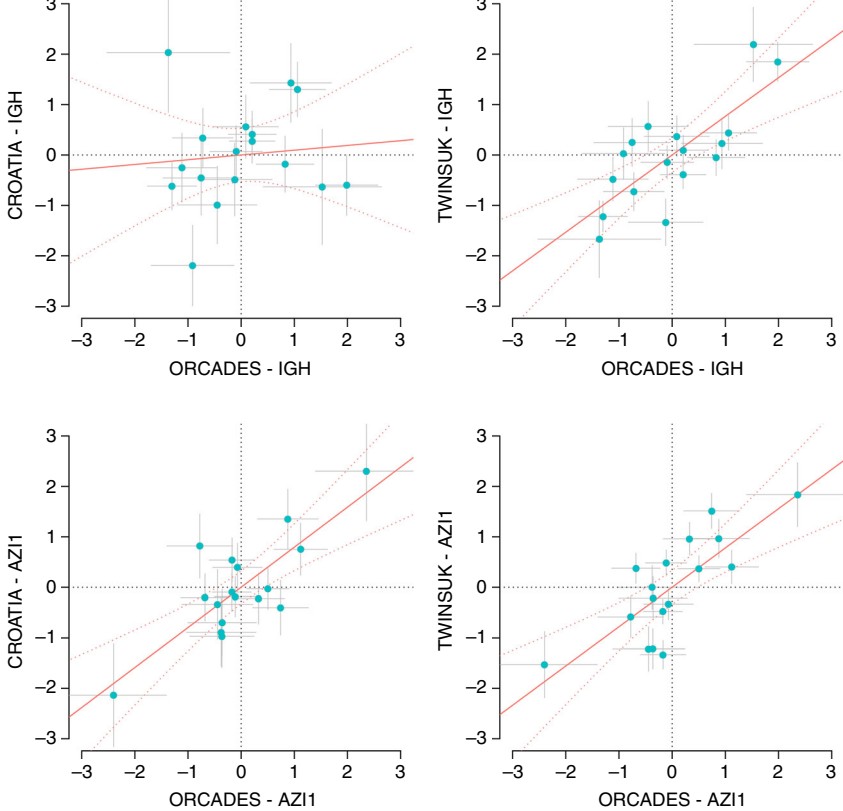

**Fig. 2** Comparison of the estimated genotype–phenotype partial correlations between discovery and replication cohorts. The *IGH* and *AZI1* loci for the 17 traits in the IgG galactosylation group are displayed. The partial correlations were standardized to *z*-scores so that the regression slope and confidence intervals (*red curves*) represent the amount of correlation in the effect sizes between each named cohort. The *gray bars* represent standard errors

replication, in which for novel loci that were not detectable by univariate GWAS (*ELL2, HLA-B-C, AZI1, FUT6-3*), we tested association between the top SNP and all 23 traits using conventional univariate model. We considered the results to be replicated by this approach if we observed an association, which was consistent between replication and discovery and with *P*-value passing Bonferroni correction for the number of SNPs and traits tested (5 SNPs and 23 traits). The results are presented in Supplementary Data 1. On the top of significant and consistent replication for all four loci considered, one can see that for *ELL2* and *FUT6* we can achieve replication of association between SNP and the "top" associated trait from univariate analysis and for *HLA-B-C* and *AZI1* we achieve replication at the second-best trait. This provides additional and convincing evidence that the reported associations are true, and suggests the score replication procedure is valid.

**Locus-specific pleiotropic model can vary across cohorts.** Geometrically, the procedure for estimating the phenotype score projects the SNP dosage vector onto the hyperplane defined by the phenotype vectors in the discovery cohort (Supplementary Fig. 10). Based on the same hyperplane of phenotypes, an exact replication would yield a projected SNP dosage vector in the replication cohort which has a correlation of 1 with that of the discovery cohort, i.e., in Supplementary Fig. 10, $\theta$ should be nearly zero. Having a significant positive genetic effect on the phenotype score in the replication cohort only replicates the fact that $|\theta| < 90°$. Namely, the "score replication" above replicates the genetic effect on the same phenotype score but does not guarantee that all the *partial* correlations between the SNP dosage and the phenotypes are consistent. Therefore, for each newly discovered locus, we tested the correlation between the set of these

partial correlations ($\rho$) in the discovery cohort to that in each replication cohort. In Table 1 and Supplementary Table 6, $r_\rho$ and $P_\rho$ represent the estimates and significance of such correlations.

The partial correlations between each genotype dosage and the phenotypes can be viewed—up to a constant—as the partial coefficients from multiple regression of the dosage on the phenotypes (shown in Supplementary Data 1). The $r_\rho$ values are rather high (all > 0.74) when comparing ORCADES and TWIN-SUK, whereas when contrasting ORCADES and the Croatians, for *HLA*, *FUT6* and some phenotype groups of *IGH*, such correlation estimates appear to be close to zero. As an example, for the IgG galactosylation phenotypes, we visualize the partial correlation contrast for the *IGH* and *AZI1* loci (Fig. 2). As we standardized the partial correlations in each cohort to *z*-scores, the regression slope in each panel represents $r_\rho$. We can see that the effects of the *AZI1* locus correlate well across replication cohorts, whereas for the *IGH* locus, the correlation is low for ORCADES vs. Croatian cohorts. From Table 1 and Supplementary Table 6, one can see this is not an exclusive feature of the galactosylation group, but can be observed for other trait groups as well. Because differences in experimental/sample collection procedures between the cohorts would not be locus-specific, they are unlikely to be the cause. We may speculate that the observation of different multivariate association patterns across populations has a genetic explanation, such as different LD structures between British and Croatian populations and/or presence of specific environmental factors modulating the action of the loci in question.

**Connection to immune-related tissues and disease.** We used DEPICT software[30] to perform gene prioritization, gene set and tissue enrichment analyses. Eighteen analyses (GWAS of 9 trait

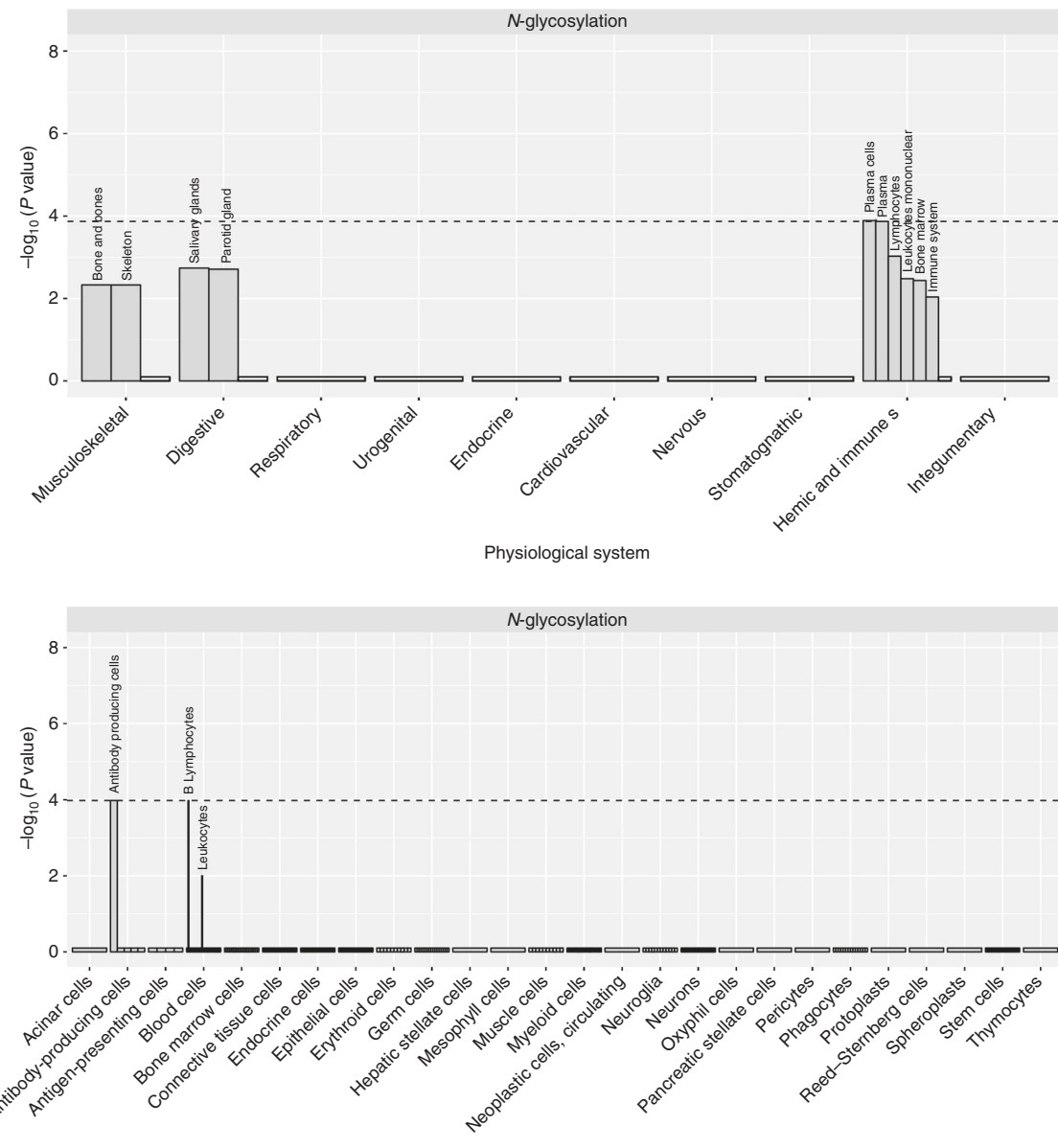

**Fig. 3** DEPICT enrichment analysis results for IgG N-glycosylation loci. (**a**) Tissues. (**b**) Cell types. Genes in the associated loci are highly expressed in cells and structures of the hemic and immune systems (with emphasis on antibody-producing cells), and to a lesser degree in the skeleton and glands of the digestive system. See Supplementary Data 2 for details

sets, using either genome-wide significant SNPs or SNPs with MANOVA $P < 1 \times 10^{-5}$) were run. Results are reported in Supplementary Data 2. The gene sets demonstrating most significant enrichment included regulation of protein kinase activity and Endoplasmic Reticulum-nucleus signaling pathway. The top enriched tissues/cell types included B-lymphocytes, plasma cells, antibody-producing cells, plasma (see Fig. 3 and Supplemetary Fig. 11–12 for trait group-specific results).

To link to established knowledge and to provide insights regarding potential underlying functions of the IgG glycosylation loci, we searched association databases using PhenoScanner (http://www.phenoscanner.medschl.cam.ac.uk/) for our reported top variants (8 and 6 for the new and known loci, respectively) and obtained 2011 association records. Filtered based on FDR < 5%, we identified associations between SNPs detected in this study and 17 complex diseases and disease-related traits (Fig. 4, Supplementary Table 5).

## Discussion

We start with discussion of the methodological implications, and continue to discuss biological aspects of our findings.

Our analyses are of methodological interest over and above the new biological insight they have provided. The fact that we discover and replicate a number of new loci using a multivariate approach is consistent with results previously demonstrated by Inouye et al.[13] for metabolomics, and, although to a lesser extent, by Stephens[10] for classical lipids. As we have demonstrated, the *P*-value adjustment method TATES did not show the same power as our multivariate analysis. In principle, TATES produces quite similar results to the univariate GWAS, because it does not jointly model the correlations among the phenotypes and genotypes, but instead adjusts the univariate GWAS results accounting for the correlations of univariate analyses statistics. However, phenotypic correlation and correlation between statistics are not necessarily consistent with each other, especially at a single genetic variant.

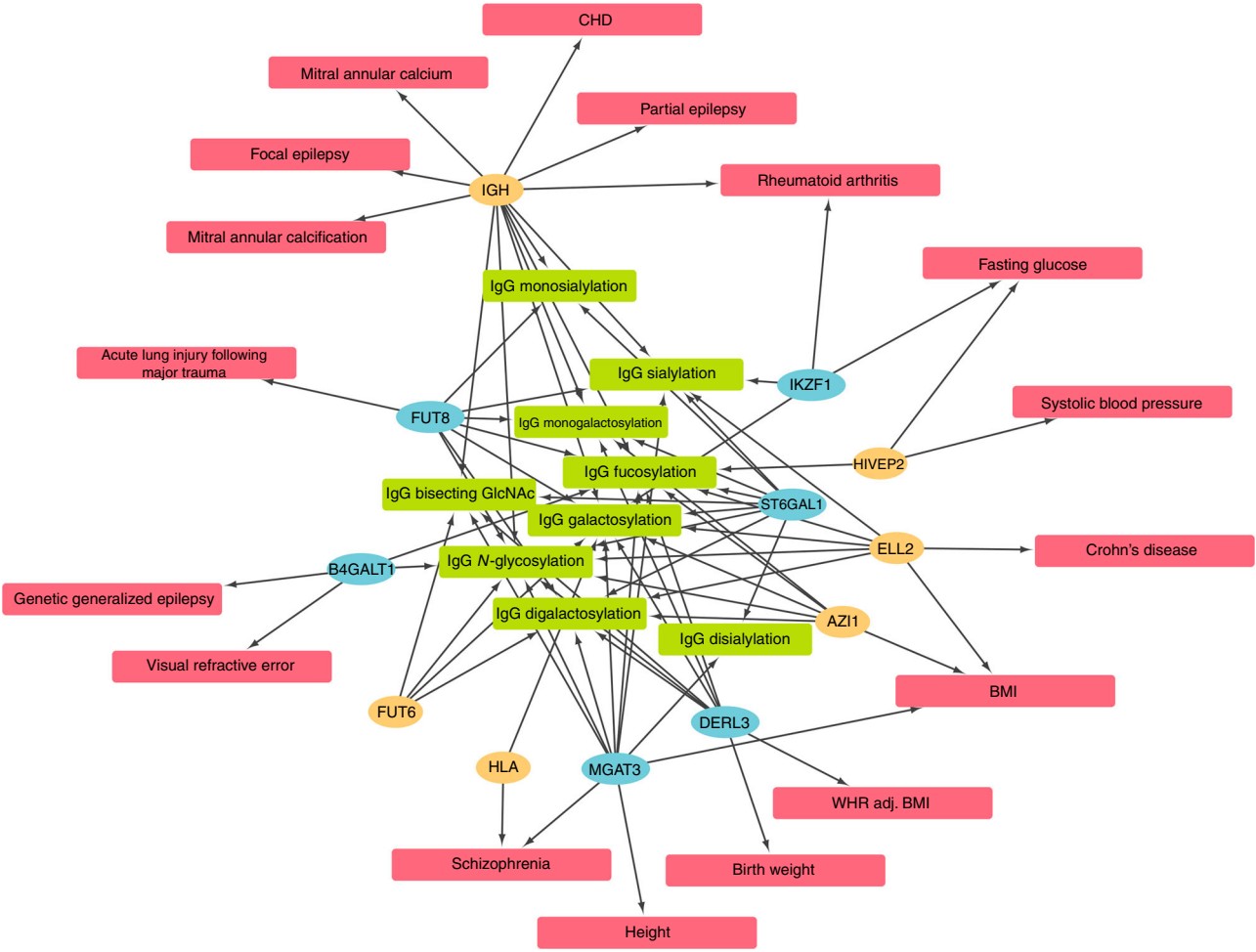

**Fig. 4** Pleiotropic network of the novel (*orange*) and known (*blue*) loci discovered using the multivariate method. The IgG *N*-glycosylation phenotypes are shown in *light green* and complex traits/disease in *red*. The associations with complex traits were filtered based on an FDR cutoff of 5%. CAD, coronary heart disease; BMI, body mass index; WHR adj BMI, waist–hip ratio adjusted for BMI

As such, TATES is not an ideal method for multivariate analysis, but rather one for multiple testing correction.

To examine the discovery power of different multi-trait methods, we contrasted the power of our efficient MANOVA statistic with GEMMA (as a representitive for mixed model based methods) and TATES (as a representative for *P*-value correction methods) under different scenarios, where we see that, for unrelated individuals, the MANOVA statistic performs no worse than the other methods (Supplementary Fig. 7). Although GEMMA should perform better than our two-step approach, the difference is only expected to become substantial when the sample includes a large group of highly related individuals, which is not the case with our discovery cohort. Because the variance-covariance structure of the phenotypes is modeled explicitly, we found that MANOVA and GEMMA methods are no worse than univariate or TATES in all scenarios investigated, and in some scenarios they substantially outperform TATES (Supplementary Fig. 7). This highlights the importance of joint modeling of multiple omic phenotypes, which is able to identify loci that were missed in univariate GWAS.

Intuitively, the power of multivariate analysis should depend on the amount and pattern of correlations among the phenotypes, and between the phenotypes and genotypes. MANOVA in general has good power for intermediately correlated phenotypes, as it approximates Fisher's method for independent phenotypes, while nearly identical phenotypes are not expected to contribute additional information. Beyond mathematical correlation, biological relevance of the phenotypes should be considered in practice, for example to link genetic variants to (sub)groups of omics measurements reflecting specific biological processes, such as the glycosylation groups in this study.

Certainly, increasing the number of tested phenotype groups would lead to more multiplicity of the statistical testing procedure. Here, we focus on nine particular groups and Bonferroni correct the number of GWA scans, in order to be conservative in reporting discoveries and follow the conventional rules in such genetic research. However, we would also emphasize that multiplicity is not simply a problem that reduces power, but also offers an opportunity to discover more.

Another perspective of looking at power increase via multivariate analysis is that the power is gained by reducing noise in the measurements when combining multiple correlated phenotypes. For example, in our replication procedure, the effect at a single locus is interpreted by expressing the genotype dosage as a linear combination of multiple phenotypes, which is similar to our common knowledge that summing up multiple repeated measurements of a trait would lead to the reduction of relative residual variance and therefore improve statistical power[31]. Given this, multivariate analysis can be applied, not only to omics data,

but also to any correlated phenotypes in general, where power may be gained due to this statistical property.

Methods for meta-analysis are critical in current statistical genomics: they allow for pooling of results of analyses from different studies thus achieving large sample sizes, statistical power and replication. As MANOVA was applied for the multivariate GWA analyses, there was no single effect size estimate for multiple phenotypes available. Meta-analysis based on the estimated effects size would therefore not be straightforward. In a large meta-analysis, we can combine the P-values with a weight for each cohort depending on its sample size. At the same time, pooled estimates of the coefficients for constructing the phenotype score could be obtained, though the meta-analysis based on such multiple coefficients will be less powerful. Meta-analysis in a replication procedure can be easily conducted based on the one degree-of-freedom test on the phenotype score estimated in the discovery cohort. However, our results suggest that pleiotropic models may differ between cohorts, which calls for extra caution (e.g., use of correlation between partial regression coefficients) when performing multivariate replication.

Our multi-phenotype method identified five novel loci associated with the human IgG N-glycome; all of them were convincingly replicated. These loci include such positional candidates as the immunoglobulin heavy locus (IGH), IgH transcription elongation factor ELL2, human leukocyte antigen (HLA-B-C), and a locus including fucosyltransferases 6 and 3 (FUT6-FUT3); additionally, we have identified a locus on chromosome 17 flanked by AZI1 and TMEM105 (Table 1, Supplementary Figs. 2–6).

The novel chromosome 19 association includes the FUT3-5-6 gene cluster (Supplementary Fig. 6). This locus was recently shown to be associated with glycosylation of plasma proteins[21]; based on our results, this is the second known locus shared in common between total plasma protein and IgG glycosylation. Products of this gene cluster are three fucosyltransferases, enzymes that catalyze transfer of fucose from the donor guanosine-diphosphate fucose to the acceptor molecules. The main products of these genes are the Lewis x and Lewis a structures that determine Lewis blood groups. However, according to current knowledge, these enzymes catalyze fucosylation of antennary GlcNAc[32], resulting in glycan structures that are not found on IgG, and therefore the mechanism through which this locus affects IgG glycosylation is not obvious.

The newly established locus on chromosome 6 includes the human leukocyte antigen class I HLA-C and HLA-B genes. The HLA super-locus on chromosome 6 is a gene-rich region that has been associated with more than a hundred, mostly autoimmune, diseases. There are at least 132 protein-coding genes, as well as the classical HLA genes. Many products of these genes are molecules involved in either innate immunity or the adaptive immune response—such as the classical HLA which encode the antigen-presentation apparatus[33].

Several potential candidates are found at the novel chromosome 17 locus. We note that this locus was suggestively associated in the previous univariate analysis of Lauc et al.[23], which included our discovery cohort ORCADES as one of the five cohorts analyzed. The peak of association is flanked by the AZI1 and TMEM105 genes, and includes several other genes, among which SLC38A10 was prioritized by DEPICT. Potentially of greater interest might be IKZF3, which encodes a transcription factor that interacts with IKZF1. The IKZF1 locus was associated with IgG glycosylation in the previous GWAS and suggestively rediscovered in this study. These two transcription factors are involved in regulation of differentiation and proliferation of B lymphocytes[34], the cells where immunoglobulins are synthesized.

For the IGH locus (Supplementary Fig. 4), no gene was prioritized by DEPICT. The locus contains genes encoding the heavy chains of immunoglobulins, also including immunoglobulin G (IGHG genes). Immunoglobulin G consists of two biologically different regions—the antigen binding fragment (Fab) and crystallizable fragment (Fc). While the Fab region is responsible for binding with antigens, the Fc region is responsible for binding with effector molecules and cells[35], guiding the immune response. Both regions can be glycosylated, with the majority of glycans coming from the Fc region and leading the immune response[36]. While a biological relationship between IGHG and IgG is obvious, it is not immediately clear what mechanism could link variation in IGHG region and IgG glycosylation. Similarly, it is interesting that in the novel associated interval on chromosome 5, the ELL2 gene is prioritized by DEPICT. ELL2 encodes the RNA polymerase II transcription elongation factor, which plays a role in immunoglobulin secretion. ELL2 regulates exon skipping of IGH and is necessary for processing mRNA transcribed from IGH[37].

In a previous GWAS, Lauc et al.[23] used single-trait analyses to detect nine loci, of which four contained genes encoding glycosyltransferases with obvious links to IgG N-glycosylation. Here, we find an additional five loci, of which only one contains a gene directly involved in protein glycosylation. Our results show that genetic control of IgG glycosylation is a complex process involving multiple biological pathways. Another interesting observation is that we observe clear biological links between some of the positional candidate genes, for example IGHG and ELL2, IKZF1 and IKZF3. With future larger studies, we should be able to further illuminate the complexity of the genetic control of glycosylation.

## Methods

**The ORCADES discovery cohort.** The Orkney Complex Disease Study (ORCADES) is a family-based study of 2078 individuals aged 16–100 years recruited between 2005 and 2011 in the isolated Scottish archipelago of Orkney[26]. Genetic diversity in this population is decreased compared to Mainland Scotland, consistent with the high levels of endogamy historically. Fasting blood samples were collected and over 300 health-related phenotypes and environmental exposures were measured in each individual. Genome-wide genotyping was performed using Illumina HumanHap300 and OmniExpress arrays.

**The KORCULA and VIS replication cohorts.** The CROATIA-Vis study includes 1008 Croatians, aged 18–93 years, who were recruited from the villages of Vis and Komiža on the Dalmatian island of Vis during 2003 and 2004 within a larger genetic epidemiology program[38]. The CROATIA-Korcula study includes 969 Croatians between the ages of 18 and 98[39]. The field work was performed in 2007 and 2008 in the eastern part of the island, targeting healthy volunteers from the town of Korčula and the villages of Lumbarda, Žrnovo and Račišće. Genome-wide genotyping was performed using Illumina HumanHap300 and OmniExpress arrays.

**The TWINSUK replication cohort.** The TwinsUK cohort (www.twinsuk.ac.uk, also referred to as the UK Adult Twin Register) is an adult twin British registry shown to be representative of the United Kingdom female population[40, 41]. From this registry, a total of 4479 subjects, had N-linked IgG glycans measurements and were included in the analysis. Genotyping was performed using the HumanHap300 and the HumanHap610Q array.

**Ethics statement.** All research in this study that involved human participants has been approved by Research Ethics Committees—in Orkney and Aberdeen for the Orkney Complex Disease Study (ORCADES); in Croatia and Edinburgh for the VIS and KORCULA studies; and by St Thomas' hospital Research Ethics Committee for TWINSUK. All ethics approvals were given in compliance with the Declaration of Helsinki (World Medical Association, 2000). All human subjects included in this study have signed appropriate written informed consent.

**Glycomic and genomic data.** Information about IgG glycosylation phenotypes can be found in Supplementary Table 2, and more details are given in Lauc et al. (2013)[23]. All samples were imputed to the 1000 Genomes using the b37 reference

panel. Variants with minor allele frequencies (MAF) < 0.05, or imputation R-squared < 0.50, were excluded from the genome scan.

**Multi-trait association test statistic.** For $k$ phenotypes, where $k$ is often much less than the sample size $n$, the association between the group of $k$ phenotypes $\mathbf{Y}_{n \times k}$ and a biallelic marker $\mathbf{g}$ can be expressed as a multivariate regression

$$\mathbf{Y}_{n \times k} = \mathbf{1}_{n \times 1} \boldsymbol{\mu}'_{k \times 1} + \mathbf{g}_{n \times 1} \boldsymbol{\beta}'_{k \times 1} + \mathbf{e}_{n \times k}$$

which can be tested via MANOVA for the null hypothesis

$$H_0 : \boldsymbol{\beta} = 0$$

Although MANOVA has been a standard multivariate test method, here, we show how a Pillai trace statistic can be obtained from the data. Note that different MANOVA test statistics are equivalent for a single marker test (Supplementary Fig. 8). Here, each column of the phenotype matrix has been GRAMMAR+ transformed[27] so that the population structure is corrected using linear mixed models (see the next subsection).

We can calculate residual variance-covariance matrix of the above multivariate linear regression as

$$\mathbf{E} = \left( \mathbf{Y} - \mathbf{1} \hat{\boldsymbol{\mu}}' - \mathbf{g} \hat{\boldsymbol{\beta}}' \right)' \left( \mathbf{Y} - \mathbf{1} \hat{\boldsymbol{\mu}}' - \mathbf{g} \hat{\boldsymbol{\beta}}' \right)$$

The corresponding residual variance-covariance matrix of the null model is

$$\mathbf{E}_0 = (\mathbf{Y} - E[\mathbf{Y}])'(\mathbf{Y} - E[\mathbf{Y}])$$

The model variance-covariance matrix captured by the genetic variant is then

$$\mathbf{H} = \mathbf{E}_0 - \mathbf{E}$$

Analog of the univariate ANOVA F-test, let $\lambda_j$ ($j = 1, \ldots, k$) be the eigenvalues solving

$$\det(\mathbf{H} - \lambda \mathbf{E}) = 0$$

Pillai's trace can be calculated as

$$\mathbf{V} = tr\left( \mathbf{H}(\mathbf{H} + \mathbf{E})^{-1} \right) = \sum_{i=1}^{k} \frac{\lambda_i}{1 + \lambda_i}$$

and the corresponding F-statistic is

$$\frac{V/k}{(1 - V)/(n - k - 1)} \sim F_{k, n-k-1}$$

When $n$ is large, $k$ times the F-statistic is approximately $\chi^2(k)$-distributed.

**Genome-wide association analyses.** Prior to GWAS, each trait was adjusted for fixed effects of sex, age, and the other experimental factors. Glycans are quantified on 96-well plates, where the plate factor represents plate membership for each sample. The column factor represents the column on the plate (twelve columns per plate) for each sample and machine the UPLC machine on which the sample was ran. While the whole plate goes through the same procedure from the first step of the experiment, the UPLC instrument can quantify only one third of the plate at time, represented in the part factor. The residuals were inverse-Gaussian-transformed to standard normal distributions. The residuals expressed as Z-scores were used for all association analyses. In both the genotypes from SNP array and 1000 Genomes-imputed data, markers with minor allele frequency < 0.05 or imputation R-square < 0.30 were excluded. GRAMMAR+ transformation[27] was implemented in the GenABEL-package[42], part of the GenABEL-suite[43]. The genomic relationship matrix used in the analyses was generated by the ibs() function (with weight = "freq" option), which uses SNP array data to estimate the realized pairwise kinship coefficient. The polygenic() function was used to obtain the GRAMMAR+ transformed phenotypes (grresidualY) from linear mixed models. All univariate GWAS inflation factors (lambda values) were close to 1, and the multivariate GWAS inflation factor was 1.005, showing that this method efficiently accounts for family structure.

We implement the above multivariate analysis in the Multivariate() function of the MultiABEL package. Implemented in this function is the MANOVA of multiple phenotypes against each single variant genotype dosage, and for the method option, in our analysis, Pillai's trace[44] was used as the test statistic for the multivariate association. Univariate analysis of each phenotype at each locus was performed using linear regression via the lm() function in R, with subsequent genomic control to correct the inflation factor. The resulted univariate P-values were passed onto TATES for P-value adjustment analysis.

**Phenotype score estimation and replication.** In the discovery cohort, the top variant genotype dosage was regressed on the multiple phenotypes, and the estimated coefficients were used for constructing the compound phenotype which is a linear combination of the original phenotypes. It should be noted that the F-test statistic of this regression model is equivalent to the MANOVA test statistic used in our GWAS. In each replication population, the same compound phenotype was constructed using the coefficients estimated in the discovery population, and thereafter, tested against the genotype dosage of the same variant using linear

regression. The effect of the variant on the compound phenotype estimated from this regression model is denoted as $\beta_s$. The R-squared from regressing the phenotype score on the genotype is equivalent to that from regressing the genotype on multiple phenotypes[45] (e.g., in the MultiPhen method, see Supplementary Fig. 8 for equivalence of statistics). Therefore, given that the joint genetic effects are homogeneous in different cohorts, the replication power using such a phenotype score is consistent with MANOVA test replication. After applying inverse Gaussian transformation, the regression coefficients represent genotype-phenotype partial correlations, which can be compared between discovery and replication cohorts as a strong replication for homogeneity of genetic effects.

**Code availability.** The free and open source R package MultiABEL is available at: https://cran.r-project.org/package=MultiABEL. Its developer version is available at the GenABEL project repository: https://r-forge.r-project.org/R/?group_id=505. Tutorial of the multivariate GWA analysis procedure using MultiABEL is available at: https://github.com/xiashen/MultiABEL/.

**Data availability.** The results of the nine genome-wide multivariate scans in the discovery analysis are available in the DataShare repository (http://dx.doi.org/10.7488/ds/2069) of the University of Edinburgh. The remaining data are contained within the paper and Supplementary Files or available from the corresponding authors upon request.

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

## Acknowledgements

We thank Chris S. Haley, Yudi Pawitan and Jeanine Houwing-Duistermaat for useful discussion of this manuscript. This study was supported by by European Commission FP7 grants MIMOmics (contract #305280), HTP-GlycoMet (contract #324400), Pai-nOmics (contract #602736) and Integra-Life (contract #315997), and H2020 projects GlySign (contract #722095), SYSCID (contract #733100) and IMforFuture (contract #721815) as well as funding for the Croatian National Centre of Research Excellence in Personalized Healthcare. X.S. was in receipt of a Swedish Research Council grant (No. 537-2014-371). The work of Y.S.A. was supported by the Federal Agency of Scientific Organisations via the Institute of Cytology and Genetics (project # 0324-2016-0008). The work of S.S. was supported by Russian Ministry of Science and Education under 5–100 Excellence Programme. ORCADES was supported by the Chief Scientist Office of the Scottish Government (CZB/4/276, CZB/4/710), the Royal Society, the MRC Human Genetics Unit, Arthritis Research UK and the European Union framework program 6 EUROSPAN project (contract no. LSHG-CT-2006-018947). DNA extractions were performed at the Edinburgh Clinical Research Facility. The CROATIA-Vis and CROATIA-Korcula acknowledge the University of Split and Zagreb Medical Schools, the Institute for Anthropological Research in Zagreb and the Croatian Institute for Public Health. Both studies were funded by grants from the Medical Research Council (UK), MSES grants (216-1080315-0302) and the Croatian Science Foundation (8875). The TwinsUK study was funded by the Wellcome Trust; the National Institute for Health Research (NIHR)-funded BioResource, Clinical Research Facility and Biomedical Research Centre based at Guy's and St Thomas' NHS Foundation Trust in partnership with King's College London.

## Author contributions

Initiated and coordinated the study: X.S. and Y.S.A. Developed statistical methods: X.S. and Y.S.A. Analyzed the data: X.S. Investigated and interpreted discoveries: X.S., L.K., S.S. and D.W. Contributed to replication analysis: M.M. Contributed to simulation studies: Z.N. Contributed discovery data: J.F.W. Contributed replication data: I.R., O.P., C.H. and T.D.S. Contributed IgG glycomics data: I.T.-A., M.P.-B., G.L. Contributed to writing: X.S., L.K., M.M., T.D.S., J.F.W., G.L. and Y.S.A.

## Additional information

**Competing interests:** The authors declare no competing financial interests. G.L. has multiple patents in the field of glycoscience issued. Y.S.A. is a director and co-owner of Maatschap PolyOmica, which provides (consulting) services in the area of (statistical) (gen)omics.

