## [Peer Review file · Nature Communications]

Reviewers' comments:

Reviewer #1 (Remarks to the Author):

This paper presents a 'multi-phenotype GWAS workflow' and applies it to 23 glycan traits, demonstrating improvement in both power (relative to two standard univariate approaches) and interpretation. The GWAS workflow primarily concatenates Gen/MultiABEL functions published elsewhere, and the only novelty is a phenotype score constructed from weights learned from a discovery/training cohort. The GWAS, which is more successful than univariate GWAS, is presented as a case study for this new workflow and for multi-phenotype analyses more generally; moreover, much exposition is dedicated to demystifying the power and apparent complexity of multi-phenotype analyses, which is a timely, important and well-delivered message. The extensive replication analysis convincingly corroborates the five novel loci.

A. The claim that some methods cited can not handle population stratification or perform powerful and meaningful replication analysis is false. As the authors surely know, conditioning on PCA scores is the standard way of accounting for population structure, and in the majority of GWAS this is fine. Both multiphen and snptest can handle covariates. In addition, the authors should cite [1-4] and discuss these approaches. They are all highly relevant.

B. Another major problem is a lack of comparison to state-of-the-art multitrait techniques; for example, gemma/limix [5,6] ``handle population stratification' (such approaches are declared nonexistent in lines 67-68). This is important because the univariate versions of these programs [7,8] outperform the univariate version of GRAMMAR in some settings. Moreover, these methods can fit covariates jointly with SNPs, which may improve power relative to adjusting for covariates as a preprocessing step. Certainly, these incur increased computational cost, but none of the studied datasets are obviously too big (with the possible exception of the all-23-phenotypes analyses). A related, but less serious, concern is the absence of alternative MANOVA test statistics, e.g. according to the paper, MQFAM uses Wilk's lambda instead of Pillai's trace.

C. Why was the ORCADES cohort used as the discovery cohort? It would have made more sense to use 1 cohort as the replication cohort, and the others jointly as discovery cohorts. What happens if the cohorts are analyzed in a different order i.e. use the largest TwinsUK cohort as discovery? Or put ORCADES, VIS and KORCULA together as one discovery cohort. If the authors claim that their method can handle any level of ancestry/kinship then this could be more powerful. These options should be explored, and will be of great interest to the field.

D. The authors have not adequately described how this study relates to their previous GWAS of glycosylation [9,10]. The authors could do a better job in the introduction of clarifying the differences in the assays/phenotypes. Also, FUT6 is found in [9,10]. Also, given that [9] claims that HNF1 is a master regulator of plasma protein fucosylation, why has this not been found or replicated in this study?

E. A drawback of the workflow is that authors have grouped phenotypes, tested the groups, and then used a conservative correction to correct for the highly correlated tests. This is clearly not optimal and should be discussed.

F. A non-standard method of replication has been used i.e creating a weighting across phenotypes which is then tested in another cohort. This should be compared to applying the same MANOVA test in each replication cohort. If the authors are suggesting this as an alternative method, then they need to validate the method. A simulation study would also be very valuable. Other researchers will want to understand which conditions influence the performance of each approach. I would also use the method in [1] to do a full meta-analysis.

G. The paper which proposed TATES reported that it was generally more powerful than MANOVA

except in specific scenarios. It is not clear whether the results presented here are in line with these findings. The authors should comment on this.

H. It is stated in the text that MultiPhen is mathematically equivalent to a multivariate analysis of variance (MANOVA). If this is true it is not obvious. I think it should be justified or referenced.

[1] MultiMeta: an R package for meta-analyzing multi-phenotype genome-wide association studies

<http://bioinformatics.oxfordjournals.org/content/early/2015/05/15/bioinformatics.btv222.full>

[2] A unified framework for association analysis with multiple related phenotypes.

<http://dx.plos.org/10.1371/journal.pone.0065245>

[3] A multiple-phenotype imputation method for genetic studies

Nature Genetics, (2016). doi:10.1038/ng.3513

[4] Efficient multiple-trait association and estimation of genetic correlation using the matrix-variate linear mixed model.

<http://www.genetics.org/content/200/1/59.full>

[5] Efficient multivariate linear mixed model algorithms for genome-wide association studies.

<http://www.nature.com/doi/10.1038/nmeth.2848>

[6] Efficient set tests for the genetic analysis of correlated traits.

<http://www.nature.com/doi/10.1038/nmeth.3439>

[7] FaST linear mixed models for genome-wide association studies.

<http://www.nature.com/doi/10.1038/nmeth.1681>

[8] Genome-wide efficient mixed-model analysis for association studies.

<http://www.nature.com/doi/10.1038/ng.2310>

[9] Polymorphisms in B3GAT1, SLC9A9 and MGAT5 are associated with variation within the human plasma N-glycome of 3533 European adults.

<http://www.hmg.oxfordjournals.org/cgi/doi/10.1093/hmg/ddr414>

[10] Genomics meets glycomics-the first GWAS study of human N-Glycome identifies HNF1 α as a master regulator of plasma protein fucosylation.

<http://dx.plos.org/10.1371/journal.pgen.1001256>

Reviewer #2 (Remarks to the Author):

This manuscript by Shen et al describes the association of several genetic regions in the human genome that can influence the N-glycosylation phenotypes of human IgG. Initial findings made in one cohort were replicated in independent cohorts, ultimately leading to the identification of five novel loci that associate with this phenotype. The authors conclude that the multivariate analysis employed in large-scale association studies will have great potential in revealing complex biology underlying high-throughput omics data.

Comments:

The manuscript does not contain functional data and mainly describes the results of an approach linking "glycomics" data with GWAs data. As I am not a statistical expert, I find it hard to judge whether the approaches used are valid and appropriate.

However, the approach taken resembles previous studies from the same group linking the human plasma N-glycome to GWAs results (see e.g.: <http://www.ncbi.nlm.nih.gov/pubmed/21908519?dopt=Abstract&holding =npg>; Hum Mol Genet. 2011 Dec 15;20(24):5000-11) and might not be state-of-art given recently published methodologies allowing multi-phenotype association studies for genetic studies (see e.g. Dahl et al. Nature Genetics 48, 466, 2016).

The finding that genetic background might influence the N-glycosylation phenotype has been made before (Lauc et al. Plos genet. 9 e1003225, 2013) and is, as such, not novel. The current manuscript replicates these original findings and presents the discovery of five novel loci. However, as no functional back-up is provided to the findings presented, the biological significance of the results is not clear and is mainly speculative at this stage. In my opinion, the impact of the data presented would increase in case some functional insight is provided to the observations presented. Likewise, the impact of the manuscript would increase in case more genetic fine-mapping would be performed, for example by determining the haplotype-boundaries/haplotype blocks that associate with N-glycosylation phenotypes of human IgG.

POINT-TO-POINT RESPONSE TO THE REVIEWERS

Re Reviewer #1 (Remarks to the Author):

*This paper presents a 'multi-phenotype GWAS workflow' and applies it to 23 glycan traits, demonstrating improvement in both power (relative to two standard univariate approaches) and interpretation. **The GWAS workflow primarily concatenates Gen/MultiABEL functions published elsewhere, and the only novelty is a phenotype score constructed from weights learned from a discovery/training cohort.***

This is a misunderstanding of the novelty of this paper. The MultiABEL functions were NOT published elsewhere, and this is our first manuscript that developed the new functions, applied them to real data analysis, discovered novel loci, and conducted thorough replications. As the reviewer points out, the phenotype score is only one part of the new method. The other important novelty is in the application of multivariate methods to real 'omics' data, and demonstration of a large increase in power for gene discovery.

In order to avoid such misunderstanding, in the revised manuscript, we provide more detailed description of the statistical model and hypothesis testing, implemented in the MultiABEL package, and provide a short tutorial for genome-wide association analysis using the new tool.

The GWAS, which is more successful than univariate GWAS, is presented as a case study for this new workflow and for multi-phenotype analyses more generally; moreover, much exposition is dedicated to demystifying the power and apparent complexity of multi-phenotype analyses, which is a timely, important and well-delivered message. The extensive replication analysis convincingly corroborates the five novel loci.

*A. The claim that **some methods cited can not handle population stratification or perform powerful and meaningful replication analysis is false.** As the authors surely know, conditioning on PCA scores is the standard way of accounting for population structure, and in the majority of GWAS this is fine. Both multiphen and snptest can handle covariates. In addition, the authors should cite [1-4] and discuss these approaches. They are all highly relevant.*

We agree with the reviewer that our argument sounded too strong without sufficient explanation. PCA scores as covariates can indeed handle population structure, but only to a certain extent. For complicated relatedness in the population, a linear mixed model-type analysis can become necessary. In the revised manuscript, we describe the possibility of including PCA scores in e.g. the PLINK multiphen procedure, and GRAMMAR+ mixed model residuals as one way of dealing with stratification. We also thank the reviewer for pointing out relevant literature. In the revised manuscript, we provide a table summarizing the functionalities, similarities and differences, and pros and cons among relevant tools (**Supplementary Table 5**).

*B. Another major problem is a **lack of comparison to state-of-the-art multitrait techniques; for example, gemma/limix** [5,6] ``handle population stratification' (such approaches are declared nonexistent in lines 67-68). This is important because the univariate versions of these programs [7,8] outperform the univariate version of GRAMMAR in some settings. Moreover, these methods can fit covariates jointly with SNPs, which may improve power relative to adjusting for covariates as a preprocessing step. Certainly, these incur increased computational cost, but none of the studied datasets are obviously too big (with the possible exception of the all-23-phenotypes analyses). A related, but less serious,*

concern is **the absence of alternative MANOVA test statistics**, e.g. according to the paper, MQFAM uses Wilk's lambda instead of Pillai's trace.

We thank the reviewer for pointing out the relevance of other techniques, although we must stress that we are dealing with and are interested in demonstrating the potential of multivariate analysis on real data (IgG glycosylation), which - as the reviewer rightly notices - restrict our choice to the faster methods (e.g. GEMMA could not be applied). We agree that GRAMMAR types of analyses can be outperformed in particular settings, and we also think that this can be generalized to multi-trait scenarios. Nevertheless, a major contribution of our paper is to show that our particular way of performing multi-trait analysis is sufficiently powerful to identify new loci for IgG glycosylation, which is a main focus of our line of glycosylation research, and results obtained allow for replications in independent populations. Despite this, in **Supplementary Figure 7** of the revised manuscript, we examine the power of our MANOVA statistic and the relevant methods GEMMA and TATES under different scenarios, where we see that, for unrelated individuals, our efficient MANOVA statistic performs no worse than the other methods. Although we believe GEMMA may perform better in more highly related individuals, our discovery cohort ORCADES does not contain too strong relatedness to cause severe power loss (standardized genomic kinship values: mean = -0.0004, sd = 0.0078, 1% quantile = -0.0063, 99% quantile = 0.0090). We therefore think that our analysis, although not 100% perfect in terms of methodology, fits our purpose of this IgG glycosylation genetic study. Moreover, with larger numbers of traits (e.g. bigger 'omics', such as metabolomics) application of theoretically superior methods such as GEMMA will become impossible, while our proposed methodology scales well with increasing numbers of traits, as no large multivariate mixed model needs to be fitted.

We found that, as variance-covariance structure of the phenotypes is modeled, MANOVA and GEMMA methods substantially outperform the single-trait p-value adjustment method TATES, indicating that the necessity of jointly analyzing multiple correlated phenotypes as single-trait analysis in TATES loses too much power. This is another very important message of this work.

For single-marker analysis, all these MANOVA test statistics are equivalent, i.e. they all correspond to the same F-statistic. To show this clearly, we add a simulation to compare four different types of MANOVA statistics (**Supplementary Fig. 8**). Therefore, the choice of particular MANOVA test statistic is not important.

C. Why was the ORCADES cohort used as the discovery cohort? It would have made more sense to use 1 cohort as the replication cohort, and the others jointly as discovery cohorts. What happens if the cohorts are analyzed in a different order ie. use the largest TwinsUK cohort as discovery? Or put ORCADES, VIS and KORCULA together as one discovery cohort. If the authors claim that their method can handle any level of ancestry/kinship then this could be more powerful. These options should be explored, and will be of great interest to the field.

We agree with the reviewer that it may be interesting to explore these different options; however, for this study we have decided to use a standard GWAS workflow in which a set of discovery and replication cohorts are defined *a priori* in order to guarantee robustness of findings and avoid cryptic multiple testing. Our multivariate analyses was based on individual level data, and therefore we used the ORCADES cohort, to which we had direct access, as the discovery cohort. We have contacted TwinsUK and Croatian cohorts which performed replication of our findings using their data; no individual-level data were exchanged at this point. We did not swap the order of discovery and replication cohorts, as clear design of discovery and replication cohorts from the beginning of the study make the process more compelling and the statistical findings more interpretable. This is a practically reasonable and

statistically more convincing pipeline, allowing for two rounds of replication, which is stronger than a single replication or a big meta-analysis without replication.

In the revised manuscript, partially following the reviewer's suggestion in (F), we combine all these cohorts and report a meta-analysis result for the novel loci, to also provide joint genetic effects estimates with small standard errors. However, we do not perform a meta-analysis of the whole genome as this is not the design of our study. A meta-analysis of these studies would be an option at a later stage, when we could guarantee availability of additional replication cohorts.

D. The authors have not adequately described how this study relates to their previous GWAS of glycosylation [9,10]. The authors could do a better job in the introduction of clarifying the differences in the assays/phenotypes. Also, FUT6 is found in [9,10]. Also, given that [9] claims that HNF1 is a master regulator of plasma protein fucosylation, why has this not been found or replicated in this study?

We thank the reviewer for pointing out the lack of description of the phenotypes and the link to previous findings. In the revised manuscript, we provide the link to previous studies in more detail in Introduction, and **Figure 2** partially shows the link to previous discoveries and their pleiotropic effect with other complex traits/disease.

With regard to [9, 10] we would like to stress that aforementioned studies are genome-wide association studies of N-glycans linked to **total plasma proteins**. In these studies glycans are released from all glycosylated plasma proteins and analysed simultaneously, resulting in a single glycan profile. In such studies it is impossible to distinguish whether observed changes in glycan profile are caused by a change of a specific glycan structure or by a change in concentration of a protein that carries specific glycan structure (in which case we would not expect the observed change to be glycosylation-driven, but rather driven by the protein itself).

Therefore, in this study we focus on glycosylation of a single protein - immunoglobulin G. In such studies, the protein is first isolated from other plasma proteins, followed by release and quantification of the glycans. In this case, changes observed in glycosylation profiles can be attributed to glycosylation rather than the protein concentration.

Considering HNF1-alpha as a major regulator of plasma protein fucosylation, we would like to stress that majority of plasma glycoproteins are synthesised in the liver (Clerc et al. 2015 Glycoconjugate Journal), where HNF1-alpha is also well expressed (Odom et al, 2014 Science). On the other hand, IgGs are synthesised in B-lymphocytes, where HNF1-alpha is not expressed.

To summarize, while some overlap in genetic control of total plasma and IgG glycome may be expected, these are two very different sets of traits. Comparing genetic control between these two may be an interesting topic to work on; however, this is beyond the scope of this work. We now explain these points in the introduction.

E. A drawback of the workflow is that authors have grouped phenotypes, tested the groups, and then used a conservative correction to correct for the highly correlated tests. This is clearly not optimal and should be discussed.

This is certainly true. We have chosen a conservative way to report our discoveries, in order to avoid false positives and only report signals with great confidence. In the revised manuscript, we discuss this concern as a new paragraph in Discussion.

F. A non-standard method of replication has been used i.e creating a weighting across phenotypes which is then tested in another cohort. This should be compared to applying the same MANOVA test in each replication cohort. **If the authors are suggesting this as an alternative method, then they need to validate the method.** A simulation study would also be very valuable. Other researchers will want to understand which conditions influence the performance of each approach. I would also use the method in [1] to do a full meta-analysis.

We thank the reviewer for making this comment. In the revision, first of all, we added the “naive” MANOVA replication to **Table 1** for comparison. In the revised Methods section we give more explanation to our other replication methods. In particular, the main reason why the phenotype score construction is chosen to conduct the replication analysis is that it provides a meaningful genetic effect on a single measurable score with specific effect direction. Simply applying the MANOVA test in replication can give only p-value replication, which does NOT guarantee that the genetic effect goes the same direction as the discovery cohort thus is of little value in genetics, nor in general science. In the revised manuscript, we also take meaningful replication to the next level, i.e. to contrast partial genotype-phenotype correlations between discovery and replication cohorts, which double-checks the consistency of genetic effects in replication analysis, not only using meaningless p-values. Therefore, we believe that the phenotype score construction is a stronger replication than the replication using MANOVA.

In the Methods section, we also clarify that mathematically, the R-squared of regressing the phenotype score on the genotype is equivalent to that of regressing the genotype on multiple phenotypes (e.g. in MultiPhen). Therefore, given that the joint genetic effects are homogeneous in different cohorts, the replication power using such a phenotype score is consistent with MANOVA test replication. However, when the joint genetic effects are heterogeneous in different cohorts, replication using the score is more difficult - whichever loci can be replicated using the score indicate that the genetic effects across multiple phenotypes are all very similar. One can replicate using the MANOVA test p-value, but replication with only p-values does not mean replication of the ‘direction’ of the genetic effects.

In the revised manuscript, we provide more explanations so that our choice of replication method makes clear sense.

G. The paper which proposed TATES reported that it was generally more powerful than MANOVA except in specific scenarios. It is not clear whether the results presented here are in line with these findings. The authors should comment on this.

We thank the reviewer for pointing this out. Our real data application indicate that TATES does not have sufficient power compared to the MANOVA, therefore we do not agree with the TATES conclusion and believe that their conclusion must be drawn under certain special scenarios that favor their method. This is explained by the fact that essentially TATES does not perform multivariate modelling, but rather multiple testing correction method.

In the revised manuscript, we add a comparison via simulation (**Supplementary Fig. 7**) and more comments. With these simple bivariate scenarios, we can see that the MANOVA method (as a classic multivariate test) has an overall performance no worse than TATES. For certain scenarios, such as 1. some phenotypes serve as correlated but non-genetic factors, or 2. genetic effects go in different directions to the phenotypic correlation, MANOVA has substantial power advantage over TATES, indicating that properly modeling phenotypic correlation is statistically more solid than simply adjusting p-values. The latter takes no consideration of the effects directions, therefore is not optimal.

H. It is stated in the text that MultiPhen is mathematically equivalent to a multivariate analysis of variance (MANOVA). If this is true it is not obvious. I think it should be justified or referenced.

We thank the reviewer for pointing out that this is not so obvious. In fact, this is the equivalence between MANOVA on the single factor and the corresponding multiple regression (see <http://arxiv.org/abs/1504.06006>). Similarly to above, in the revised manuscript, we added a simulation to show this identity (**Supplementary Fig. 8**).

[1] MultiMeta: an R package for meta-analyzing multi-phenotype genome-wide association studies.

<http://bioinformatics.oxfordjournals.org/content/early/2015/05/15/bioinformatics.btv222.full>

[2] A unified framework for association analysis with multiple related phenotypes.

<http://dx.plos.org/10.1371/journal.pone.0065245>

[3] A multiple-phenotype imputation method for genetic studies.

Nature Genetics, (2016). doi:10.1038/ng.3513

[4] Efficient multiple-trait association and estimation of genetic correlation using the matrix-variate linear mixed model.

<http://www.genetics.org/content/200/1/59.full>

[5] Efficient multivariate linear mixed model algorithms for genome-wide association studies.

<http://www.nature.com/doifinder/10.1038/nmeth.2848>

[6] Efficient set tests for the genetic analysis of correlated traits.

<http://www.nature.com/doifinder/10.1038/nmeth.3439>

[7] FaST linear mixed models for genome-wide association studies.

<http://www.nature.com/doifinder/10.1038/nmeth.1681>

[8] Genome-wide efficient mixed-model analysis for association studies.

<http://www.nature.com/doifinder/10.1038/ng.2310>

[9] Polymorphisms in B3GAT1, SLC9A9 and MGAT5 are associated with variation within the human plasma N-glycome of 3533 European adults.

<http://www.hmg.oxfordjournals.org/cgi/doi/10.1093/hmg/ddr414>

[10] Genomics meets glycomics-the first GWAS study of human N-Glycome identifies HNF1 α as a master regulator of plasma protein fucosylation.

<http://dx.plos.org/10.1371/journal.pgen.1001256>

Re Reviewer #2 (Remarks to the Author):

This manuscript by Shen et al describes the association of several genetic regions in the human genome that can influence the N-glycosylation phenotypes of human IgG. Initial findings made in one cohort were replicated in independent cohorts, ultimately leading to the identification of five novel loci that associate with this phenotype. The authors conclude that the multivariate analysis employed in large-scale association studies will have great potential in revealing complex biology underlying high-throughput omics data.

Comments:

The manuscript does not contain functional data and mainly describes the results of an approach linking "glycomics" data with GWAs data. As I am not a statistical expert, I find it hard to judge whether the approaches used are valid and appropriate.

However, the approach taken resembles previous studies from the same group linking the human plasma N-glycome to GWAs results (see e.g.:

*<http://www.ncbi.nlm.nih.gov/pubmed/21908519?dopt=Abstract&holding=npg>; Hum Mol Genet. 2011 Dec 15;20(24):5000-11) and **might not be state-of-art given recently published methodologies allowing multi-phenotype association studies for genetic studies** (see e.g. Dahl et al. Nature Genetics 48, 466, 2016).*

As the reviewer points out, we were indeed not providing functional data, as the main goals of this work were 1. to develop a multi-phenotype GWA pipeline that corrects for any sort of population structure and 2. to investigate how much power is gained by application of multivariate method to real 'omics' data and consequently 3. to discover, convincingly replicate, and report novel loci for IgG glycosylation. Nevertheless, in the revision, we performed additional *in silico* functional follow-ups (see below and answers to subsequent comments).

There is a misunderstanding in terms of the approach taken. Indeed our earlier work, as the reviewer mentioned, was also a GWAS, but, the statistical approach developed and applied in this paper is novel, and has never been applied to glycomic data before. This is a multivariate analysis that is very different from conventional GWA analysis. There are indeed multiple groups in the world working on multi-phenotype analysis methods, because such methods are strongly needed - this shows that the topic is indeed a state-of-art scientific area. Specifically, regarding the Dahl et al. paper, it does **NOT** have the same aim in terms of methods as our paper, as they focus on imputation of the missing phenotypes rather than multivariate genome-wide association studies. In fact, combining Dahl et al.'s method and our development will be a strong tool for many data applications, but this is outside of scope of work reported here.

In the revised manuscript, we provide further analysis linking our findings to established knowledge in various database, in order to provide insights regarding the underlying functions of the novel loci. First of all, we searched association databases (<http://www.phenoscanter.medschl.cam.ac.uk/>) for our reported top variants and obtained 653 association records. Filtered based on FDR < 5%, we identified associations between our novel loci and complex disease or disease-related traits (**Figure 3, Supplementary Table 7**). These may indicate shared genetic basis of IgG N-glycosylation and various complex traits.

*The finding that **genetic background might influence the N-glycosylation** phenotype has been made before (Lauc et al. Plos genet. 9 e1003225, 2013) and **is, as such, not novel.***

This is a misunderstanding, and we are sorry we were not clear enough. We do not conclude that genetic background (polygenic effects) influences IgG glycosylation, rather we are

correcting for this, so that we find real causal loci that influence IgG glycans, regardless of population structure.

*The current manuscript replicates these original findings and presents the discovery of five novel loci. However, as **no functional back-up is provided to the findings presented, the biological significance of the results is not clear and is mainly speculative at this stage.** In my opinion, the impact of the data presented would increase in case some functional insight is provided to the observations presented. Likewise, the impact of the manuscript would increase in case more genetic fine-mapping would be performed, for example by determining the haplotype-boundaries/haplotype blocks that associate with N-glycosylation phenotypes of human IgG.*

To elucidate function of loci associated with IgG glycosylation we performed Data-driven Expression Prioritized Integration for Complex Traits (DEPICT). This method accumulates information from diverse sources, from manually curated gene co-expression and protein-protein interaction networks to various pathway databases (such as REACTOME and KEGG) to uncover biological pathways enriched for glycosylation associated loci and to identify tissues and cells where these genes are highly expressed (**Figure 3**). While (somewhat) expectedly we see loci we discover are enriched for genes expressed in antibody-producing cells, this not only works as proof-of-principle for discovered loci, but also this finding is clearly worth documenting for use by other researchers. In particular, this finding reinforces our decision to use B-cells as the system for further functional follow-up studies.

In order to characterize the molecular consequences of the novel variants associated with IgG N-glycosylation, we examined all the genotyped and imputed variants with multi-trait p-values less than 10^{-4} at the five novel loci using Ensembl variant effect predictor (http://www.ensembl.org/Homo_sapiens/Tools/VEP). As expected, most variants reside in noncoding regions, where about 2%-8% of the variants at each locus are within regulatory regions (**Supplementary Fig. 9**).

REVIEWERS' COMMENTS:

Reviewer #2 (Remarks to the Author):

In my original report, I mentioned three issues that diminished by enthusiasm for the manuscript:

1. Impact; . The manuscript does not contain functional data and mainly describes the results of an approach linking "glycomics" data with GWAs data and "no functional back-up is provided to the findings presented; the biological significance of the results is not clear and is mainly speculative at this stage"

2. Novelty: "the approach taken resembles previous studies from the same group linking the human plasma N-glycome to GWAs results; "The finding that genetic background might influence the N-glycosylation phenotype has been made before (Lauc et al. Plos genet. 9 e1003225, 2013) and is, as such, not novel"

3. genetic fine mapping is lacking "genetic fine-mapping would performed, for example by determining the haplotype-boundaries/haplotype blocks that associate with N-glycosylation phenotypes of human IgG"

In my opinion none of these comments have been addressed in the revised version of the manuscript in a convincing manner:

Ad 1) Although an in silico analysis has now been performed, no functional data are provided as also acknowledged by the authors. Therefore, the functional consequences of the findings presented remain speculative.

Ad 2 and 3) The authors indicate that, indeed, a GWAs study was presented in earlier work, but that the analyses performed this time was novel due to new statistical approaches. I would consider this a relative minor increment over existing data without additional functional biological significance or genetic fine mapping.

Reviewer #3 (Remarks to the Author):

The revised version of the paper did an excellent job addressing the previous reviews. Overall, the paper presents a success story from multivariate GWAS analysis which shows a replication. This in itself is very important.

There are a few places where the manuscript can be clearer. The method describes an approach very similar to MANOVA. It would help to elaborate in both the main text and in the Methods what exactly are the differences between what is applied and what is in other implementations of MANOVA. Perhaps it would make sense to have an overview of what is novel in the statistical method in the results section.

The authors added some analysis comparing to other methods. This is currently only mentioned in the discussion. I think the authors should bring it earlier when discussing the relationship to other methods.

POINT-TO-POINT RESPONSE TO THE REVIEWERS' COMMENTS:

Reviewer #2 (Remarks to the Author):

In my original report, I mentioned three issues that diminished by enthusiasm for the manuscript:

1. Impact; . The manuscript does not contain functional data and mainly describes the results of an approach linking "glycomics" data with GWAs data and "no functional back-up is provided to the findings presented; the biological significance of the results is not clear and is mainly speculative at this stage"

2. Novelty: "the approach taken resembles previous studies from the same group linking the human plasma N-glycome to GWAs results; "The finding that genetic background might influence the N-glycosylation phenotype has been made before (Lauc et al. Plos genet. 9 e1003225, 2013) and is, as such, not novel"

3. genetic fine mapping is lacking "genetic fine-mapping would performed, for example by determining the haplotype-boundaries/haplotype blocks that associate with N-glycosylation phenotypes of human IgG"

In my opinion none of these comments have been addressed in the revised version of the manuscript in a convincing manner:

Ad 1) Although an in silico analysis has now been performed, no functional data are provided as also acknowledged by the authors. Therefore, the functional consequences of the findings presented remain speculative.

We agree that the functional consequences of findings presented remain only a hypothesis that we have generated in the process of our work, and that will have to be tested in the future.

Ad 2 and 3) The authors indicate that, indeed, a GWAs study was presented in earlier work, but that the analyses performed this time was novel due to new statistical approaches. I would consider this a relative minor increment over existing data without additional functional biological significance or genetic fine mapping.

We see the value of our work not in generating the new data, but in developing new data analysis tools and extracting new genetic knowledge from previously generated data. We have developed new powerful data analysis procedures, new approaches to replication and applied these to achieve

convincing results, and finally discovered five new loci (on the top of only 9 known) for human IgG *N*-glycosylation. Larger, even more powerful, analyses, development of even better data and technological tools, fine mapping and molecular biological investigation in a wet lab will be important future contributors into our progress towards better understanding of genetic and biological control of human IgG *N*-glycosylation.

Reviewer #3 (Remarks to the Author):

The revised version of the paper did an excellent job addressing the previous reviews. Overall, the paper presents a success story from multivariate GWAS analysis which shows a replication. This in itself is very important.

We really appreciate the referee's understanding of the core novelty of our work!

There are a few places where the manuscript can be clearer. The method describes an approach very similar to MANOVA. It would help to elaborate in both the main text and in the Methods what exactly are the differences between what is applied and what is in other implementations of MANOVA. Perhaps it would make sense to have an overview of what is novel in the statistical method in the results section.

MANOVA test is essential part of our procedure. Our novelty, besides the biological discoveries and strong replication, is to introduce transformation of multiple phenotypes based on linear mixed models prior to the MANOVA test, so that the population structure can be properly corrected for the multivariate analysis.

In the revised manuscript, we emphasize in the Methods (lines 427-429) that the phenotypes need to be corrected for population stratification before applying any MANOVA test. We also bring this to Results (lines 133-138) to clarify this step of our analysis, as correction for population structure is essential to genomic studies but difficult to implement for multivariate analysis.

The authors added some analysis comparing to other methods. This is currently only mentioned in the discussion. I think the authors should bring it earlier when discussing the relationship to other methods.

We agree with the referee that this is an important point to introduce early in the manuscript. Due to the editorial limitation, we have a 1000-word limit for the Introduction, nevertheless, in the revised manuscript, we create a Supplementary Note 1 and an interested reader is directed to it already in the Introduction. In this note, we demonstrate the pros and cons of different existing methods (listed in Supplementary Table 1) and their connections.